# Combining the GP's assessment and the PHQ-9 questionnaire leads to more reliable and clinically relevant diagnoses in primary care

**Clara Teusen**[1] *, **Alexander Hapfelmeier**[1,2], **Victoria von Schrottenberg**[1], **Feyza Gökce**[1], **Gabriele Pitschel-Walz**[1], **Peter Henningsen**[3], **Jochen Gensichen**[4], **Antonius Schneider**[1], **for the POKAL-Study-Group**[¶]

**1** Institute of General Practice and Health Services Research, TUM School of Medicine, Technical University of Munich, Munich, Bavaria, Germany, **2** Institute for AI and Informatics in Medicine, TUM School of Medicine, Technical University of Munich, Munich, Bavaria, Germany, **3** Dept. of Psychosomatic Medicine and Psychotherapy, University Hospital TU Munich, Munich, Bavaria, Germany, **4** Institute of General Practice and Family Medicine, University Hospital of the Ludwig-Maximilians-University of Munich, Munich, Bavaria, Germany

¶ Complete membership of the POKAL-Study-Group can be found in the Acknowledgments.
* clara.teusen@mri.tum.de

**Data Availability Statement:** Our data contain potentially identifying or sensitive patient

## Abstract

### Background

Screening questionnaires are not sufficient to improve diagnostic quality of depression in primary care. The additional consideration of the general practitioner's (GP's) assessment could improve the accuracy of depression diagnosis. The aim of this study was to examine whether the GP rating supports a reliable depression diagnosis indicated by the PHQ-9 over a period of three months.

### Methods

We performed a secondary data analysis from a previous study. PHQ-9 scores of primary care patients were collected at the time of recruitment (t1) and during a follow-up 3 months later (t2). At t1 GPs independently made a subjective assessment whether they considered the patient depressive (yes/no). Two corresponding groups with concordant and discordant PHQ-9 and GP ratings at t1 were defined. Reliability of the PHQ-9 results at t1 and t2 was assessed within these groups and within the entire sample by Cohen's Kappa, Pearson's correlation coefficient and Bland-Altman plots.

### Results

364 consecutive patients from 12 practices in the region of Upper Bavaria/Germany participated in this longitudinal study. 279 patients (76.6%) sent back the questionnaire at t2. Concordance of GP rating and PHQ-9 at t1 led to higher replicability of PHQ-9 results between t1 and t2. The reliability of PHQ-9 was higher in the concordant subgroup ($\kappa = 0.507$) compared to the discordant subgroup ($\kappa = 0.211$) ($p = 0.064$). The Bland-Altman Plot showed that the deviation of PHQ-9 scores at t1 and t2 decreased by about 15% in the concordant

information. Therefore, the Medical Ethics Committee of the Technical University Munich has restricted data access. The data are held by the Institute of General Practice and Health Services Research of the Technical University Munich. The data are not publicly available due to data protection regulations, but may be obtained from the Institute of General Medicine and Health Services Research of the Technical University Munich by researchers who meet the criteria for access to confidential data. Interested researchers can contact the data protection officer of the Technical University Munich if they wish to access our data (e-mail: beauftragter@datenschutz.tum. de). Alternatively, data requests may be sent to the Institute of General Medicine and Health Services Research of the Technical University Munich (e-mail: allgemeinmedizin@mri.tum.de).

**Funding:** This secondary data analysis was funded by the German Research Foundation (Deutsche Forschungsgesellschaft, https://www.dfg.de/) (grant No GrK 2621). As principal investigator of the graduate school „PrädiktOren und Klinische Ergebnisse bei depressiven ErkrAnkungen in der hausärztLichen Versorgung (POKAL)" AS received the funding to conduct the analysis. The funders had no role in study design, data collection and analysis, decision to publish, or preparation of the manuscript.

**Competing interests:** The authors have declared that no competing interests exist.

subgroup. Pearson's correlation coefficient between PHQ-9 scores at t1 and t2 increased significantly if the GP rating was concordant with the PHQ-9 at t1 ($r = 0.671$) compared to the discordant subgroup ($r = 0.462$) ($p = 0.044$).

## Conclusions

The combination of PHQ-9 and GP rating might improve diagnostic decision making regarding depression in general practices. PHQ-9 positive results might be more reliable and accurate, when a concordant GP rating is considered.

## Introduction

Epidemiological studies show that depression is one of the major health problems worldwide [1, 2]. Thus, the accurate diagnosis and appropriate treatment of patients with depressive disorder is a key challenge to our health system [3–5]. Often, the first point of contact for patients with depression is the general practitioner (GP) [6]. Differentiating depressive symptoms from somatic, non-specific, functional or somatoform body complaints is difficult for GPs [7], particularly if patients are multimorbid and seek help for physical rather than psychological complaints [8–10]. Furthermore, compared to specialists in mental health care, the work of GPs takes place in a setting with a high risk of both over and under diagnosis of depression due to the presence of multimorbidity [3]. Even though about 10% of primary care patients are likely to meet criteria for major depression, detection and treatment rates are still low [11]. In addition to that, the low-threshold access to patients in general practices and the confrontation with subthreshold symptoms that could indicate a multitude of possible diseases complicate the correct diagnosis of depression [12, 13]. However, a correct diagnosis is essential for further adequate treatment of depression.

Previous studies investigated how GPs deal with the challenges of diagnostic decision making regarding depression. It was shown that the GP's approach often differs from common psychiatric diagnostic systems like the International Statistical Classification of Diseases and Related Health Conditions (ICD-10) or the Diagnostic and Statistical Manual of Mental Disorders (DSM-V) [14]. According to ICD-10 and DSM-V, a depressive disorder can be diagnosed after a 2-week reference period [15]. In general practice, this time criterion is sometimes considered too short to confirm a depression diagnosis [16]. Usually, only if depressive symptoms persist for a longer period of time a depression diagnosis is considered and discussed carefully with the patient. Accordingly, GPs may be better at diagnosing correctly more severe than mild depression [17], in part because this subtype of depression is associated with a longer duration of symptoms [18]. Other studies found that the 2-week reference period for depression is appropriate and valid in general practice [19, 20]. However, in order to initiate even better diagnosis for depression in primary care, it might be helpful to take into account GP heuristics, their diagnostic strategies and thought processes in addition to existing psychiatric diagnostic criteria [3, 21–24]. A systematic review identified the "experienced anamnesis" and the patient's long-term history as decisive factors for the GP's diagnosis [25]. The length of the doctor-patient relationship is thus an important factor for GP diagnostics [26]. Besides these, other heuristics that GPs use for diagnostic decision making are watchful waiting, consideration of etiological and contextual factors and stepwise diagnostic procedures [25, 27].

A common strategy to improve diagnostic decision making is the introduction of screening questionnaires [28]. It has been shown that the systematic use of validated screening tools can

improve detection and diagnosis of depression in primary care [29]. However, the use of depression screening questionnaires in primary care is discussed controversially, as a screening strategy leads to an over-estimation of the depression prevalence [30]. For example, screening questionnaires like the Patient Health Questionnaire 9 (PHQ-9) lead to high false-positive rates due to the low pre-test probability of depression in primary care [30]. In addition, the diagnosis of depression is affected by the correlation between somatic illnesses and the somatic symptoms of depression [31, 32]. Whereas a PHQ-9 positive result with a PHQ-9 score ≥10 may represent clinically significant depressive symptoms, a structured diagnostic interview is needed to confirm the presence of major depressive disorder. Beyond that, the PHQ-9 seems to be more suitable for ruling-out than for ruling-in the diagnosis of depression in primary care [33–35]. Therefore, there are conflicting opinions about the recommendation of routinely screening for depression in primary care [36]. The Canadian Task Force on Preventive Health Care [37] and the guideline for depression management from the United Kingdom's National Institute for Health and Clinical Excellence [34] do not recommend routinely screening for depression in primary care settings whereas the US Preventive Services Task Force recommends a universal screening approach for depression in the general adult population [38]. We suppose that a simplified combination of screening questionnaires and GP heuristics could be useful to improve diagnostic quality and practicability in primary care.

The aim of the present analysis is to examine whether the GP rating of the patient as depressive is related to the persistence of depressive symptoms indicated by the PHQ-9 at the time of recruitment and during a follow up three months later. We assume that a longer duration of symptoms as determined with repeated PHQ-9 measurements is stronger associated with a depression diagnosed by the GP.

## Methods

### Study design and sample

We performed a secondary data analysis from a previous study on the impact of a complex educational intervention on diagnostic accuracy of impaired mental health in general practices [26]. 12 general practitioners in the region of Upper Bavaria/Germany agreed to take part in this longitudinal study. Suitable practices were recruited through the network of the 210 general practitioners with teaching duties at the Technical University of Munich (TUM). The data collection was carried out between March and October 2014.

All consecutive adult patients who attended the practices in the study period on certain days at regular intervals were asked in the waiting room by a research assistant to fill in a PHQ-9 questionnaire before seeing the doctor (t1). Additionally, at time point t1, the GPs made a subjective assessment whether they considered the patient depressive (yes/no). GPs were blinded to the PHQ-9 result of the patients they were supposed to assess for depression. During a follow-up investigation three months later (t2), patients were invited to fill in the PHQ-9 again to investigate the stability of the depressive symptoms. The questionnaire was sent by post and patients were asked to return it to the Institute of General Practice and Health Services Research. The patients received an incentive of 10 € per completed questionnaire. Inclusion criteria were an age of at least 18 years, sufficient knowledge of the German language and a signed consent form. Patients were not asked if they received any therapy, counselling or medication in the meantime. The underlying data for this study are pseudonymized and the study was approved by the Medical Ethics Committee of the Technical University of Munich (approval No 15/14).

## Questionnaire

The validated German version of the Patient Health Questionnaire 9 (PHQ-9) was used as a screening questionnaire to assess the presence of depressive symptoms in the patients within the past two weeks [39]. The depression severity score comprises nine items which can be summarized, with a range from zero (no depression) to 27 (maximum). Findings from previous studies show that the use of a cut-off value of 10 or higher is considered useful [40] as a score of 10 represents at least a moderate level of depressive symptoms [39]. A score between five and 10 is mostly found in patients with mild or subthreshold depressive symptoms and corresponds to a mild degree of severity [39]. Further on, the questionnaire presented to the patients at t1 comprised sociodemographic items regarding education, occupation and family status.

## Data analysis

The distribution of quantitative data is described by mean values and standard deviations. Qualitative data is presented by absolute and relative frequencies. Statistical significance of respective group differences was assessed by Chi-Squared Tests and t-Tests. The main research question was whether patients' self-ratings on the PHQ-9 at t1 and t2 were more reliable when the GP's rating matched with the PHQ-9 assessment at t1, compared to when there was no match. For this purpose, a PHQ-9 $\geq$10 was used to indicate a self-rated depression. This outcome was labelled as PHQ-9 positive, PHQ-9 <10 was labelled as PHQ-9 negative. Two groups with concordant versus discordant PHQ-9 and GP ratings at t1 were defined. The replicability of PHQ-9 results between t1 and t2 was compared between the concordant and discordant group by a 2 x 2 table. Mean values were presented additionally to enable a comparison of the dimensional PHQ-9 results. Furthermore, reliability of the PHQ-9 test results at t1 and t2 was assessed within these groups and within the entire sample by Cohen's Kappa, Pearson's correlation coefficient and Bland-Altman plots. Limits of Agreement (LoA) which cover about 95% of the differences between measurements were computed for the latter. Respective hypothesis testing of group differences was performed by Z-tests. All analyses were performed using the software package SPSS (Version 25, IBM, Armonk, NY, USA) and R 4.0.3 (The R Foundation for Statistical Computing, Vienna, Austria). Exploratory two-sided 5% significance levels were used for any hypothesis testing.

Within the previous cluster randomised controlled pilot study the 12 practices were divided into a control and an intervention group after a one-time training intervention for general practitioners [26]. The GPs in the intervention group received a one-day training on diagnostics and interviewing as well as on recognising and dealing with psychosomatic patients. The training included expert lectures (on depression, anxiety and somatization), group discussions and acting out psychosomatic counselling situations with an acting patient. However, a one-day training intervention alone did not seem to improve the perception and management of psychosomatic illness [26]. No relevant sociodemographic or diagnostic differences were found between the intervention and the control group, so that collected data of all patients were merged in our secondary analysis.

## Results

364 consecutive patients from 12 practices in the region of Upper Bavaria/Germany participated in this longitudinal study (see Fig 1). 279 patients (76.6%) sent back the questionnaire at t2. Non-responder analysis showed no relevant differences with respect to gender and depression diagnosis by the GP or PHQ-9 (not in Table 1). However, the average age of non-responders (mean 47.7, standard deviation 19.0) was significantly (p = 0.043) lower compared

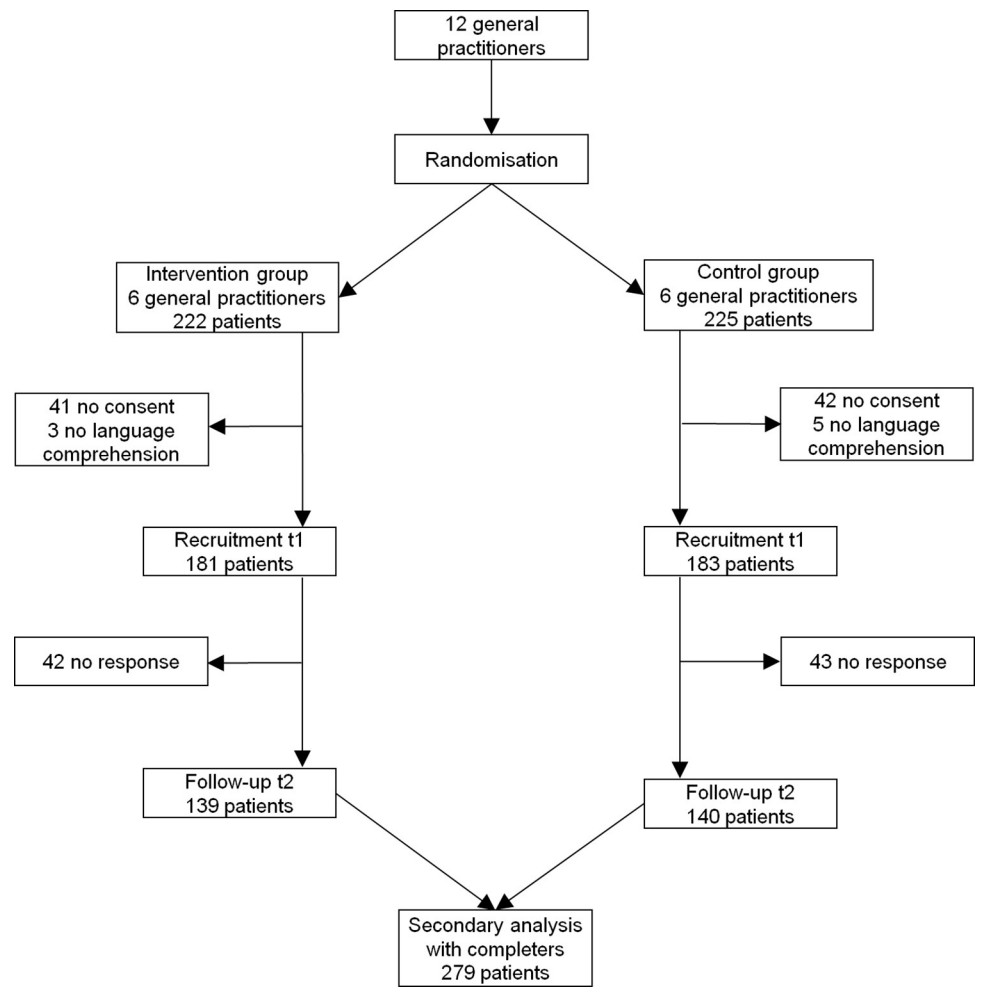

**Fig 1. Flowchart of patients.**

to responders (mean 52.3, standard deviation 18.1). The baseline characteristics of the patients at t1 are displayed in Table 1.

Table 2 depicts the replicability of the PHQ-9 results over the period from t1 to t2. Related to the entire sample, 47 patients (17.9%) were PHQ-9 positive at t1; and 216 patients (82.1%) were PHQ-9 negative. 25 (53.2%) of the PHQ-9 positives received a positive result at t2 again. 194 patients (89.8%) without depression at t1 remained without depression at t2.

The PHQ-9 test result in terms of inclusion or exclusion of depression was concordant with the GP rating at t1 in 208 patients. In this concordant subgroup, 27 patients (13.0%) received a PHQ-9 positive result at t1. 17 of these patients (63.0%) were PHQ-9 positive at t2 again. 181 patients (87.0%) were PHQ-9 negative at t1; and 166 patients (91.7%) were PHQ-9 negative at t2 again.

The discordant group with a mismatch between PHQ-9 and GP assessment at t1 comprised 55 patients. In this subgroup, 20 patients (36.4%) were PHQ-9 positive at t1. 8 of these patients (40.0%) received this positive result at t2 again. 35 patients (63.6%) were PHQ-9 negative at t1; and 28 patients (80.0%) were PHQ-9 negative at t2 again. The PHQ-9 mean values were higher in the concordant subgroup compared to the discordant subgroup in case of PHQ-9 positives

**Table 1. Baseline characteristics (t1).**

| Parameter (Missing values) | Total (n = 364) n(%) | Depression rating by the GP[1] | | | Depression at t1 (PHQ-9 ≥10)[2] | | |
|---|---|---|---|---|---|---|---|
| | | Yes (n = 85) n(%) | No (n = 271) n(%) | P-Value | Yes (n = 61) n(%) | No (n = 290) n(%) | P-Value |
| Sex, female (0) | 203 (55.8) | 55 (64.7) | 142 (52.4) | p = 0.046 | 38 (62.3) | 159 (54.8) | p = 0.285 |
| Age (6) [Mean (SD)] in years | 51.24 (18.41) | 50.98 (18.56) | 51.16 (18.23) | p = 0.803 | 45.82 (16.95) | 51.90 (18.61) | p = 0.158 |
| Marital status (2) | | | | p = 0.087 | | | p<0.001 |
| Married or in a stable relationship | 245 (67.3) | 50 (58.8) | 193 (71.2) | | 27 (44.3) | 210 (72.4) | |
| Single | 92 (25.3) | 29 (34.1) | 61 (11.4) | | 27 (44.3) | 60 (20.7) | |
| Widowed | 25 (6.9) | 6 (7.1) | 17 (6.3) | | 6 (9.8) | 19 (6.6) | |
| Education level (2) | | | | p = 0.485 | | | p = 0.659 |
| No school degree | 6 (1.6) | 3 (3.5) | 3 (1.1) | | 2 (3.3) | 4 (1.4) | |
| < 10 y of formal education | 107 (29.4) | 26 (30.6) | 79 (29.2) | | 15 (24.6) | 83 (28.6) | |
| 10 y of formal education | 112 (30.8) | 28 (32.9) | 81 (29.9) | | 22 (36.1) | 88 (30.3) | |
| 12–13 y of formal education | 116 (31.9) | 23 (27.1) | 92 (33.9) | | 17 (27.9) | 97 (33.3) | |
| Other | 21 (5.8) | 5 (5.9) | 15 (5.5) | | 4 (6.6) | 17 (5.9) | |
| Occupation (5) | | | | p = 0.017 | | | p<0.001 |
| Employed part-time | 61 (16.8) | 17 (20.0) | 43 (15.9) | | 12 (19.7) | 46 (15.9) | |
| Employed full-time | 146 (40.1) | 26 (30.6) | 119 (43.9) | | 17 (27.9) | 126 (43.4) | |
| Housewife/homemaker/non-working | 16 (4.4) | 3 (3.5) | 13 (4.8) | | 5 (8.2) | 11 (3.8) | |
| Retired | 94 (25.8) | 23 (27.1) | 68 (25.1) | | 10 (16.4) | 80 (27.6) | |
| Unemployed | 10 (2.7) | 7 (8.2) | 3 (1.1) | | 7 (11.5) | 3 (1.0) | |
| Other | 32 (8.8) | 7 (8.2) | 23 (8.5) | | 8 (13.1) | 22 (7.6) | |

[1] 8 missings due to missing GP rating

[2] 13 missings due to incomplete PHQ-9 response. SD: Standard deviation.

at t1. Likewise, the PHQ-9 mean values were lower in the concordant subgroup compared to the discordant subgroup in case of PHQ-9 negatives at t1, suggesting that PHQ-9 means are more clearly positioned above or below the cut-off value (≥10) in the concordant subgroup.

The PHQ-9 results at t1 and t2 showed a moderate agreement overall (κ = 0.430, SE = 0.072). They were higher if the GP rating was concordant with the PHQ-9 test result at t1 (κ = 0.507, SE = 0.086) compared to a fair agreement in the discordant group (κ = 0.211, SE = 0.135), still this comparison was not significant (p = 0.064) (data not in Table 2).

The Bland-Altman Plot (Fig 2) indicates that the agreement of PHQ-9 scores at t1 and t2 for the concordant subgroup is higher (Limits of Agreement (LoA): -7.25 to 7.88) than in the discordant subgroup (LoA: -9.75 to 9.33). The LoA are 95% prediction intervals describing the range in which the majority of the individual differences between the PHQ-9 measurements at t1 and t2 are expected to lie, as they cover about 95% of these values. The deviation of PHQ-9 scores at t1 and t2 decreases by about 15% when the GP rating is concordant with the PHQ-9 at t1. The plots show that patients with a PHQ-9 mean between 5 and 10 had large absolute differences between t1 and t2, suggesting high variability of the PHQ-9 scores in this range. Patients with a PHQ-9 mean <5 and a PHQ-9 mean ≥10 showed lower absolute differences between the time points, especially in the concordant subgroup.

A scatter plot regarding the correlation of PHQ-9 scores at t1 and t2 is given in Fig 3. Pearson's correlation coefficient between PHQ-9 scores at t1 and t2 was r = 0.646. Pearson's correlation coefficient increased significantly if the GP rating was concordant with the PHQ-9 at t1 (r = 0.671) compared to the discordant subgroup (r = 0.462) (p = 0.044). All correlation coefficients were statistically significant (p<0.001).

**Table 2. Replicability of the PHQ-9 results at t1 and t2 alone compared to the stability of the PHQ-9 results in the PHQ-9 and GP rating concordant vs. discordant subgroup at t1.**

| | All responders[1] (N = 263) | | | | | | |
|---|---|---|---|---|---|---|---|
| | Depression (PHQ-9≥10) at t2 | PHQ-9 at t1 | PHQ-9 at t2 | No depression (PHQ-9<10) at t2 | PHQ-9 at t1 | PHQ-9 at t2 | All |
| Depression (PHQ-9≥10) at t1 | 25 (53.2) | 15.2±4.9 | 15.2±3.5 | 22 (46.8) | 12.4±2.1 | 6.1±1.9 | 47 (17.9) |
| | | 14; 10–27 | 15; 10–25 | | 12; 10–17 | 6.7; 1–9 | |
| No depression (PHQ-9<10) at t1 | 22 (10.2) | 5.3±2.9 | 12.2±2.4 | 194 (89.8) | 3.1±2.5 | 3.3±2.2 | 216 (82.1) |
| | | 5; 0–9 | 11; 10–18 | | 2.5; 0–9 | 3; 0–9 | |
| | Concordant subgroup at t1 (N = 208) | | | | | | |
| | Depression (PHQ-9≥10) at t2 | PHQ-9 at t1 | PHQ-9 at t2 | No depression (PHQ-9<10) at t2 | PHQ-9 at t1 | PHQ-9 at t2 | All |
| Depression (PHQ-9≥10) at t1 | 17 (63.0) | 16.6±5.3 | 15.5±3.9 | 10 (37.0) | 12.3±2.3 | 6.2±1.4 | 27 (13.0) |
| | | 17; 10–27 | 15; 10–25 | | 12; 10–16 | 6.4; 4–8 | |
| No depression (PHQ-9<10) at t1 | 15 (8.3) | 4.7±2.9 | 11.8±2.2 | 166 (91.7) | 2.9±2.4 | 3.1±2.1 | 181 (87.0) |
| | | 4; 0–9 | 11; 10–16 | | 2; 0–9 | 3; 0–9 | |
| | Discordant subgroup at t1 (N = 55) | | | | | | |
| | Depression (PHQ-9≥10) at t2 | PHQ-9 at t1 | PHQ-9 at t2 | No depression (PHQ-9<10) at t2 | PHQ-9 at t1 | PHQ-9 mean t2 | All |
| Depression (PHQ-9≥10) at t1 | 8 (40.0) | 12.1±2.0 | 14.7±2.8 | 12 (60.0) | 12.5±2.0 | 6.0±2.4 | 20 (36.4) |
| | | 12; 10–16 | 14; 12–20 | | 12; 10–17 | 6.9; 1–9 | |
| No depression (PHQ-9<10) at t1 | 7 (20.0) | 6.7±2.1 | 13.1±2.8 | 28 (80.0) | 4.3±2.8 | 4.3±2.5 | 35 (63.6) |
| | | 7; 4–9 | 13; 10–18 | | 3.5; 0–9 | 4; 0–9 | |

[1]With complete PHQ-9 response; Descriptive statistics are n (%), Mean ± Standard Deviation and Median (Range).

## Discussion

The analysis showed that the concordance of the GP rating and the PHQ-9 results at t1 leads to a higher replicability of a PHQ-9 positive result over a period of three months. In addition, the replicability of a PHQ-9 negative result was improved, when GP rating and PHQ-9 were concordant at t1.

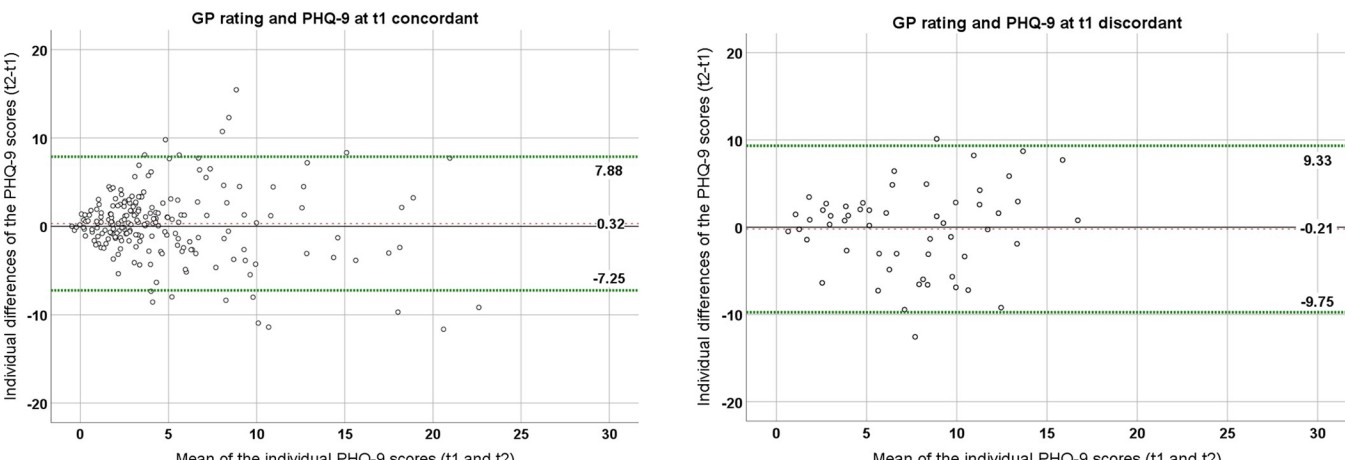

**Fig 2. Bland-Altman Plots of the subgroups with a concordant vs. discordant GP rating and PHQ-9 results at t1 (dashed lines represent the Limits of Agreement and the bias).**

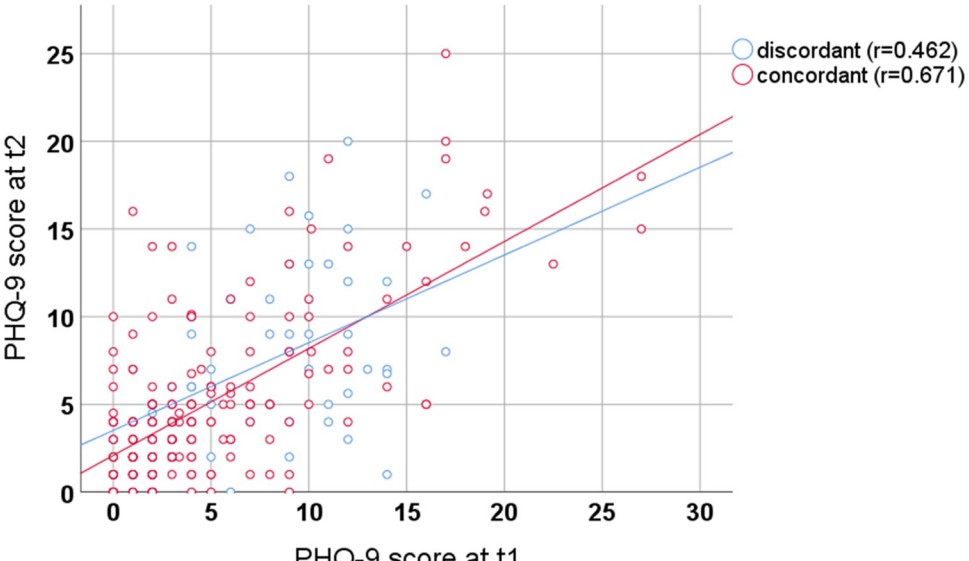

**Fig 3. Correlation of PHQ-9 scores at t1 and t2 with discordant vs. concordant PHQ-9 and GP rating.**

We found for the entire sample, that only 53.2% of all PHQ-9 positives at t1 received the same PHQ-9 positive result at t2. This proportion increased to 63.0% if the GP rating and PHQ-9 results were concordant at t1.

Beyond that, the Bland-Altman Plot showed a higher agreement of PHQ-9 scores between baseline and follow-up in the concordant subgroup. An increased reliability was also indicated by the significantly higher correlation of PHQ-9 scores between t1 and t2 in the concordant subgroup. Assuming that the persistence of symptoms is more strongly associated with actual depression [17, 18], our results suggest that a better ruling-in of depression is achieved when the GP rating in addition to a high PHQ-9 score is considered. The explicit consideration of GP heuristics like the "experienced anamnesis", the patient's long-term history [25], watchful waiting, consideration of etiological and contextual factors and stepwise diagnostic procedures [25, 27] could be of great use to increase the pre-test probability of depression in primary care. Therefore, as the mere use of screening questionnaires in general practices does not lead to sufficient diagnostic certainty [30], the combination of PHQ-9 and the GP rating might improve the detection of patients with depression in primary care.

Studies have shown that the false-positive rate of the PHQ-9 is around 60% in a population with a 10% prevalence of depression so that false-positives and false-negatives have to be examined carefully [30]. Overall we found that 46.7% of the patients with a PHQ-9 positive result at t1 had a negative result at t2 in the present study. This may indicate a significant overestimation of PHQ-9 positives at t1. In the discordant subgroup, this was even more pronounced (60.0%) than in the concordant subgroup (37.0%). Therefore, the combination of the GP and PHQ-9 assessments may be associated with a decreased likelihood of false-positives.

The accurate ruling-out of depression is also of great importance. In our analysis, we found that patients who were not rated as depressive by their GP had less PHQ-9 positive results at t1 and t2. Other studies have already shown that GPs are good at ruling-out depression [3, 21, 22]. PHQ-9 results were more reliable in the concordant subgroup, in particular when PHQ-9 scores were <5 at t1 and t2. This suggests that a better ruling-out of a depression is achieved when the GP's rating is combined with negative PHQ-9 results.

In contrast to that, patients with intermediate PHQ-9 scores (5–10) remain in a "grey area" which seems to represent a major challenge for diagnostic decision making, especially if GPs are less confident in their abilities to identify depression [24]. However, it is of great importance to increase reliable diagnostic decision making in cases with milder forms of depression which are characterized by high diagnostic uncertainty.

In the discordant subgroup, patients' PHQ-9 scores were more often just above or just below the cut-off at t1, so they were more likely to drift above or below the cut-off at t2. Recognizing stable and clear cases with pronounced depressive symptoms or without symptoms seems to be easier for GPs, while diagnosing patients with subthreshold symptoms is a major challenge. GPs are more likely to identify a more severe depression indicated by higher PHQ-9 results and are more likely to exclude a depression diagnosis correctly if the PHQ-9 result is low.

Furthermore, it has to be taken into account that among 20 patients with PHQ-9 positive results at t1 in the discordant subgroup, a significant proportion (40.0%) of patients received a positive result at t2 again, which implies the general tendency of chronicity [41, 42]. Nevertheless, there is a remarkable difference to the concordant subgroup where chronicity rates are higher (63.0%). This might indicate again, that a combination of GP heuristics and screening questionnaire could improve the diagnostic process of patients with severe depression and a high risk of chronicity, which needs to be investigated in further studies.

## Limitations

Our analysis has several limitations. First, influential life circumstances, such as therapy consultation, medication or critical life events were not measured between t1 and t2. This could have an impact on the interpretation of the results as we do not know what happened in the meantime between t1 and t2. Future studies should address this point to analyze the impact of such factors. Secondly, the results were derived by a secondary analysis. Therefore, the findings of the present study should be validated within further diagnostic studies. Thirdly, the GP rating of the patient as depressive is a subjective evaluation based on implicit GP heuristics which were used to indicate a depression diagnosis (yes/no). However, we did not identify which specific GP heuristics were used to diagnose depression. Therefore, an in-depth exploration of GP heuristics and the differentiation from psychiatric diagnostics seems to be of great value. Moreover, we could not verify whether the GP explicitly used the diagnostic criteria of depression during patient assessment. Further limitations arise from the patients who refused to participate in the study at t2. These non-responders were on average younger than responders, which might have an effect on the appearance of depression diagnoses at t2. An important limitation is given by the PHQ-9 itself. The questionnaire is well suited as a screening method for depression, but it cannot be used to obtain a reliable and definite psychological diagnosis. To be certain, such a diagnosis must be confirmed by a standardized diagnostic interview, which was not performed in this study. As it is difficult to determine the accurate proportion of false-positives and false-negatives without a reference standard, further studies need to investigate the accuracy of GP diagnoses in combination with screening questionnaires compared to standardized diagnostic interviews as a reference standard.

## Conclusion

The combination of the PHQ-9 and the GP rating might improve diagnostic decision making regarding depression in general practice. Therefore, it is necessary to combine common diagnostic methods and GP heuristics to improve the positive predictive value of screening questionnaires like the PHQ-9. Thus, a questionnaire which specifically considers the GP heuristics

as well as the psychiatric criteria might be useful. Further studies are necessary to identify explicit GP heuristics which might increase diagnostic accuracy in primary care.

## Acknowledgments

The POKAL-Study-Group (PrädiktOren und Klinische Ergebnisse bei depressiven ErkrAnkungen in der hausärztLichen Versorgung (POKAL, DFG-GRK 2621)) consists of the following principle investigators: Tobias Dreischulte, Peter Falkai, Jochen Gensichen, Peter Henningsen, Markus Bühner, Caroline Jung-Sievers, Helmut Krcmar, Karoline Lukaschek, Gabriele Pitschel-Walz and Antonius Schneider. The following doctoral students are as well members of the POKAL-Study-Group: Jochen Vukas, Puya Younesi, Feyza Gökce, Victoria von Schrottenberg, Petra Schönweger, Hannah Schillock, Jonas Raub, Philipp Reindl-Spanner, Lisa Hattenkofer, Lukas Kaupe, Carolin Haas, Julia Eder, Vita Brisnik, Constantin Brand, Katharina Biersack and Regina Wehrstedt von Nessen-Lapp. The study was performed for the PhD thesis of CT at the Medical Faculty of the Technical University Munich.

## Author Contributions

**Conceptualization:** Antonius Schneider.

**Formal analysis:** Clara Teusen, Alexander Hapfelmeier, Antonius Schneider.

**Funding acquisition:** Antonius Schneider.

**Investigation:** Clara Teusen, Alexander Hapfelmeier, Antonius Schneider.

**Methodology:** Clara Teusen, Alexander Hapfelmeier, Antonius Schneider.

**Project administration:** Antonius Schneider.

**Resources:** Antonius Schneider.

**Supervision:** Antonius Schneider.

**Writing – original draft:** Clara Teusen.

**Writing – review & editing:** Clara Teusen, Alexander Hapfelmeier, Victoria von Schrottenberg, Feyza Gökce, Gabriele Pitschel-Walz, Peter Henningsen, Jochen Gensichen, Antonius Schneider.

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
