## [Decision Letter · Decision Letter 0]

27 Jun 2022

PONE-D-21-39064Combining the GP’s assessment and the PHQ-9 questionnaire leads to more reliable and clinically relevant diagnoses in primary carePLOS ONE

Dear Dr. Teusen,

Thank you for submitting your manuscript to PLOS ONE. After careful consideration, we feel that it has merit but does not fully meet PLOS ONE’s publication criteria as it currently stands. Therefore, we invite you to submit a revised version of the manuscript that addresses the points raised during the review process.

Your manuscript has been assessed by two expert reviewers, whose comments are appended below and in the attached document. The reviewers have highlighted concerns about several aspects of the methodology and study design, among other issues. Please ensure you respond to each point carefully in your response to reviewers document, and modify your manuscript accordingly.

We look forward to receiving your revised manuscript.

Kind regards,

Joseph Donlan

Editorial Office

PLOS ONE

Journal Requirements:

Reviewers' comments:

Reviewer's Responses to Questions

**Comments to the Author**

1. Is the manuscript technically sound, and do the data support the conclusions?

Reviewer #1: Yes

Reviewer #2: No

2. Has the statistical analysis been performed appropriately and rigorously? 

Reviewer #1: Yes

Reviewer #2: No

3. Have the authors made all data underlying the findings in their manuscript fully available?

Reviewer #1: Yes

Reviewer #2: Yes

4. Is the manuscript presented in an intelligible fashion and written in standard English?

Reviewer #1: Yes

Reviewer #2: Yes

5. Review Comments to the Author

Reviewer #1: I have attached my detailed comments in the Reviewer comment documents. They are also itemized there. The paper is a useful contribution to the literature and my comments are addressable in a revised manuscript

Reviewer #2: Most psychiatric care is delivered in primary care settings, where depression is the most common presenting psychiatric symptom. Given the high prevalence of depression worldwide and the well-established consequences of untreated depression, the ability of primary care clinicians to effectively diagnose and treat it is critically important. The systematic use of validated screening tools can improve recognition and diagnosis (J Clin Psychiatry. 2020 17;81(2):UT17042BR1C.). Clinical depression diagnosis by GPs was not always associated with a formal diagnosis through a SCID (Fam Pract. 2019 Jan 25;36(1):3-11.)

.

In the background.

1. The authors mentioned that the prevalence of depression in GPs is low. In fact, previous studies in various countries have shown that the prevalence of depression in primary care is about 5-10%. The authors may state the findings of previous studies, rather than state as "low-prevalence setting (page 3).

2. A depressive disorder should be diagnosed using a 2-week reference period, which is the gold standard of ICD-10 or DSM-5 diagnostic criterion. One may argue the time criterion is too short to confirm a depression diagnosis, however, most of the studies found the ICD-10 criteria for depression seem to be appropriate and valid in general practice (Journal of Affective Disorders 2001, 65(2):191-4; J Affect Disord . 2013 Jun;148(2-3):338-46.

3. I cannot agree with the hypothesis that a 3-month duration of symptoms as determined with repeated PHQ-9 measurement might be an indication for a “real” depression. This seems to imply that a patient in primary care needs to suffer for 3 months’ watchful waiting to confirm a diagnosis.

In the method:

1. Patients were not asked if the received any therapy (counseling? psychotherapy? medication) during the three months, which would have a significant impact on the interpretation of results.

2. No psychiatric diagnostic interviews to confirm the diagnosis is a significant weakness. Gold standard (structured interview, or semi-structured interview, or a diagnosis confirmed by psychiatrist) should be used to confirm whether GP’s rating depression is under or over-diagnosed. The results of screening scale may also have problems of false negatives and false positives.

3. The results of PHQ-9 was based on scores (10-14; 15-19; 20- 27) instead of using the German version of the PHQ-9 cut-point or the gold standard for diagnostic interviews. According to a meta-analysis in 2021 (J Affect Disord. 2021 Jan 15;279:473-483.), the accuracy of the PHQ-9 was evaluated in 31 (74%) studies with a two-stage screening system, with structured interview most often carried out by primary care and mental health professionals. Most of the studies employed a cut-off score of 10 (N=24, 57%; total range 5-15). The authors may cite this paper to support why they used PHQ-9 scores (10-14; 15-19; 20- 27) instead of using the German version of the PHQ-9 cut-point. Also, they need to consider the false negatives and false positives.

In the discussion: Patients who were not rated as depression by their GP had less PHQ positive results at T1 and T2. In fact, Among 20 patients (36%) with PHQ-9 positive at T1 in the discordant group (n=55), a significant proportion (40%) of patients received a positive results at T2 again, which implies the tendency of chronicity.

6. PLOS authors have the option to publish the peer review history of their article (what does this mean?). If published, this will include your full peer review and any attached files.

Reviewer #1: No

Reviewer #2: No

---

## [Author Response · Author response to Decision Letter 0]

9 Aug 2022

Dear Editor, dear Reviewers,

Thank you for giving us the opportunity to submit a revised version of our manuscript. We are grateful for the reviewer's thoughtful comments, which have helped to improve the manuscript. We have incorporated almost all the comments in the revised version and included a point by point response to the reviewer’s comments. 

We have adapted the text/font of the Bland-Altmann plot (Fig 2) and of the scatter plot (Fig 3) and uploaded both in a modified form. We hope that the new form of the figures is more production-ready and meets the requirements of the reviewers and the journal. I would kindly ask you to contact me if anything is missing or needs to be edited.

COMMENTS

Reviewer #1

This is a useful study examining concordance between a depression measure (PHQ-9) and general practitioners’ (GP) ratings of depression. Strengths include sample size (n=364), longitudinal assessment at two time points (baseline and 3 month follow-up) and generally appropriate analyses.

I have attached my detailed comments in the Reviewer comment documents. They are also itemized there. The paper is a useful contribution to the literature and my comments are addressable in a revised manuscript

Thank you for the compliment and for highlighting the strengths of our study. We tried to address the reviewer’s helpful comments in our revised manuscript.

1. The greater reliability of the concordant group could be partly due to the possibility that those with GP depression at t1 had higher PHQ-9 scores and those with GP no depression had lower PHQ-9 scores, compared to the discordant group. Thus, if the discordant group had PHQ-9 scores closer to just above or just below 10 at t1, they would be more likely to drift above or below 10 at t2. This could be examined by showing mean PHQ-9 scores in Table 2. An example is shown below.

Thank you very much for your valuable comment. To examine whether the reliability in the concordant group was partly due to higher PHQ-9 scores when GPs indicated depression and due to lower PHQ-9 scores when the GPs indicated no depression compared to the discordant group, we have included your suggestion to show PHQ-9 means in our table (Table 2). We also included the median and range as we believe that these measures more adequately describe the distribution of PHQ-9 values and specifically their closeness to the cut-off value in the discordant subgroup. The PHQ-9 mean values and standard deviation have been inserted in the table as you suggested. We do now mention in the result section that the PHQ-9 means of the concordant subgroup are positioned more clearly above or below the cut-off: 

“The PHQ-9 mean values were higher in the concordant subgroup compared to the discordant subgroup in case of PHQ-9 positives at t1. Likewise, the PHQ-9 mean values were lower in the concordant subgroup compared to the discordant subgroup in case of PHQ-9 negatives at t1, suggesting that PHQ-9 means are more clearly positioned above or below the cut-off value (≥10) in the concordant subgroup”.

Furthermore, we highlighted in the discussion that GPs are more likely to identify a more severe depression indicated by higher PHQ-9 results and are more likely to exclude a depression diagnosis correctly if the PHQ-9 result is low. However, in the discordant subgroup, there are no such clear-cut cases, as the PHQ-9 values cluster closely around the cut-off: 

“In the discordant subgroup, patients' PHQ-9 scores were more often just above or just below the cut-off at t1, so they were more likely to drift above or below the cut-off at t2. Recognizing stable and clear cases with pronounced depressive symptoms or without symptoms seems to be easier for GPs, while diagnosing patients with subthreshold symptoms is a major challenge. GPs are more likely to identify a more severe depression indicated by higher PHQ-9 results and are more likely to exclude a depression diagnosis correctly if the PHQ-9 result is low”.

2. Table 1.

a. Define what PHQ-9 Cutoff value is, either in the column heading itself or a footnote.

b. Except for the Total column, the percentages represent row rather than column percentages. This is different from what published tables usually show and does not allow direct comparison of the “Yes” and “No” groups. For example, the proportion of women who were Yes and No for depression rating by the GP were 55/85 and 142/271, or 64.7% and 52.4%, respectively. This latter numbers directly show the proportional difference in women between the depressed and nondepressed group. The current row percentages (27.1% and 70.1%) do not show this. Thus, the authors should change percentages in table to reflect column rather than row percentages.

a. Thank you very much for this important hint. We added the PHQ-9 cut-off value ≥ 10 in the column heading of Table 1. 

b. This suggestion is very helpful. We changed the percentages in Table 1 to reflect column rather than row percentages. This makes it easier to identify the proportional differences and allows direct comparison of the “Yes” and “No” groups.

3. Were GPs blinded to PHQ-9 when they made their assessment of depression? If not (or if we don’t know) this should be added as a study limitation in the Discussion, since knowledge of the PHQ-9 could have influenced their assessment of depression (i.e., the two methods would not be entirely independent).

Thank you for raising this point. Yes, the GPs were blinded and did not know about the PHQ-9 result of the patient they were supposed to assess for depression. We now added an explanatory sentence in the methods section: 

“GPs were blinded to the PHQ-9 result of the patients they were supposed to assess for depression”.

4. Lines 81-85 are overstated: 

a. The sentence “Thus, a PHQ-9 positive result can only give a hint towards a possible depression and needs to be confirmed by a structured diagnostic interview.” There is a large amount of data supporting the construct validity of scores of 10 or greater representing clinically significant depressive symptoms. Sometimes too much is made of a “major depressive disorder diagnosis”. I might rephrase the sentence to something like: “Whereas a PHQ-9 score ≥10 may represent clinically significant depressive symptoms, a structured diagnostic interview is needed to confirm the presence of major depressive disorder.”

b. The sentence: “Therefore, the standardised and legitimised use of screening questionnaires in primary care has not yet been established.” This is an overstatement. Canadian and some European guidelines are less enthusiastic about depression screening than US guidelines and this nuance should be reflected rather than just stating a “legitimized use of screening questionnaires in primary care has not yet been established.”

a: Thank you for your suggestion. We have taken your suggestion into account and changed the sentence to: 

“Whereas a PHQ-9 positive result with a PHQ-9 score ≥10 may represent clinically significant depressive symptoms, a structured diagnostic interview is needed to confirm the presence of major depressive disorder”. 

b. Thank you for the important advice. We agree that the sentence was an overstatement and that a more detailed reflection of different approaches seems more appropriate. Now we write: 

“Therefore, there are conflicting opinions about the recommendation of routinely screening for depression in primary care [36]. The Canadian Task Force on Preventive Health Care [37] and the guideline for depression management from the United Kingdom’s National Institute for Health and Clinical Excellence [34] do not recommend routinely screening for depression in primary care settings whereas the US Preventive Services Task Force recommends a universal screening approach for depression in the general adult population [38]”.

5. Line 123 states: “A score of 10 thus indicates the presence of depression [32].” This is too simplistic. Instead, a score of 10 or greater represents a moderate level of depressive symptoms. The way the authors state it sounds like a “depression diagnosis.” Also, the reference for a cutpoint of 10 is not reference 32 (Spitzer 1999) but instead the Kroenke et al 2001 reference on the PHQ-9 in J Gen Intern Med.

We agree that the sentence is too simplistic and the diagnosis of depression should be treated with great care. We have corrected the sentence and inserted the correct reference - thank you for pointing this out. In fact, in this context, it is important to talk about the level of depressive symptoms instead of talking about a depression diagnosis. Now we write: 

“Findings from previous studies show that the use of a cut-off value of 10 or higher is considered useful [40] as a score of 10 represents at least a moderate level of depressive symptoms [39]”.

6. Line 145 – A few sentences with more detail describing the “training intervention” would be helpful since it could have affected GP diagnosis rates.

The results of the previous cluster randomized controlled pilot study show that there were no differences between the intervention and the control group. To clarify the content of the training intervention we added the following sentences in the methods section: 

“The GPs in the intervention group received a one-day training on diagnostics and interviewing as well as on recognising and dealing with psychosomatic patients. The training included expert lectures (on depression, anxiety and somatization), group discussions and acting out psychosomatic counselling situations with an acting patient. However, a one-day training intervention alone did not seem to improve the perception and management of psychosomatic illness [26]”. 

7. Lines 183-185 – Could the authors explain a little better “Limits of Agreement” – is this similar or different from 95% CI. The interpretation of Bland-Altman graphs will be unfamiliar to many readers

The limits of agreement (LoA) are a 95% prediction interval which describes the distribution of individual values as it covers about 95% of the values. By contrast, a 95% confidence interval covers an unknown population based parameter, e.g. the expectation µ which is estimated by the sample mean, with a likelihood of 95%. The idea behind the LoA is that we are interested in the agreement between the measurements at t1 and t2 for most (95%) of the patients. A 95% CI only informs us about the precision of the estimate of the mean value, which is not informative if we want to assess agreement of paired values on an individual level. To make this point more understandable for the reader, we have added the following sentence to the manuscript: 

“The LoA are 95% prediction intervals describing the range in which the majority of the individual differences between the PHQ-9 measurements at t1 and t2 are expected to lie, as they cover about 95% of these values”.

8. Minor points

a. Line 54. “Subliminal” should probably be “subthreshold”

b. Bland-Altman plots text/font, along axes, is difficult to read, and a more production-ready version of these graphs should be provided.

a. We agree. Thank you for the perceptive consideration. We corrected this.

b. Thank you for this comment. We have adapted the Bland-Altman plots and hope that they are now more in line with the expected standards.  

Reviewer #2

Most psychiatric care is delivered in primary care settings, where depression is the most common presenting psychiatric symptom. Given the high prevalence of depression worldwide and the well-established consequences of untreated depression, the ability of primary care clinicians to effectively diagnose and treat it is critically important. The systematic use of validated screening tools can improve recognition and diagnosis (J Clin Psychiatry. 2020 17;81(2):UT17042BR1C.). Clinical depression diagnosis by GPs was not always associated with a formal diagnosis through a SCID (Fam Pract. 2019 Jan 25;36(1):3-11.)

We agree that GPs play a very important role in the diagnosis and treatment of depression. In order to improve diagnostic decision making in primary care, it should be investigated whether the GP's assessment in addition to a screening questionnaire leads to better diagnostic results. If the additional GP assessment is useful, GP heuristics could be identified and included in new screening tools adapted to the primary care setting. We have added a sentence to the introduction that makes it even clearer that screening tools can lead to an improvement in diagnostics in primary care: 

“It has been shown that the systematic use of validated screening tools can improve detection and diagnosis of depression in primary care [29]”. 

In the background:

1. The authors mentioned that the prevalence of depression in GPs is low. In fact, previous studies in various countries have shown that the prevalence of depression in primary care is about 5-10%. The authors may state the findings of previous studies, rather than state as "low-prevalence setting (page 3).

Thank you very much for this valuable advice. It is true that depression is relatively prevalent in primary care compared to other diseases, at 5-10%. However, compared to the inpatient setting, this rate seems to be relatively low. With our sentence about the low-prevalence setting, we wanted to make a comparison with the inpatient setting. This comparison was possibly misleading and not clearly expressed. For this reason, we deleted the part about the low-prevalence setting and reformulated the sentence: 

“Furthermore, compared to specialists in mental health care, the work of GPs takes place in a setting with a high risk of both over and under diagnosis of depression due to the presence of multimorbidity [3]. Even though about 10% of primary care patients are likely to meet criteria for major depression, detection and treatment rates are still low [11]”.

2. A depressive disorder should be diagnosed using a 2-week reference period, which is the gold standard of ICD-10 or DSM-5 diagnostic criterion. One may argue the time criterion is too short to confirm a depression diagnosis, however, most of the studies found the ICD-10 criteria for depression seem to be appropriate and valid in general practice (Journal of Affective Disorders 2001, 65(2):191-4; J Affect Disord . 2013 Jun;148(2-3):338-46.

We have included the fact that the 2-week reference period is presented as sufficient in several studies. We believe that the additional consideration of GP heuristics can significantly improve the diagnostic process. These heuristics sometimes include watchful waiting and the 2-week criterion is exceeded. However, we added a sentence to point out that previous studies have already shown that the gold standard of a 2-week reference period works as well in primary care: 

“Other studies found that the 2-week reference period for depression is appropriate and valid in general practice [19, 20]. However, in order to initiate even better diagnosis for depression in primary care, it might be helpful to take into account GP heuristics, their diagnostic strategies and thought processes in addition to existing psychiatric diagnostic criteria”.

3. I cannot agree with the hypothesis that a 3-month duration of symptoms as determined with repeated PHQ-9 measurement might be an indication for a “real” depression. This seems to imply that a patient in primary care needs to suffer for 3 months’ watchful waiting to confirm a diagnosis.

We agree, the implication that a patient in primary care needs to suffer for 3 months’ watchful waiting to confirm a diagnosis is not correct. We would like to apologize for the slightly misleading presentation of our hypothesis. With the sentence we wanted to express that patients with a longer duration of symptoms are more likely to suffer from major depression which is more easily detected by a GP. However, this does not rule out the possibility that patients who have only had symptoms for two weeks can also suffer from major depression. To clarify any ambiguities, we have deleted the sentence in the last paragraph of the introduction section.

In the method:

4. Patients were not asked if the received any therapy (counseling? psychotherapy? medication) during the three months, which would have a significant impact on the interpretation of results.

Thank you for this comment. You are right, this is a significant limitation of our study. We have included this point in the limitations of our study already. Now we emphasized this limitation even more: 

“This could have an impact on the interpretation of the results as we do not know what happened in the meantime between t1 and t2. Future studies should address this point to analyze the impact of such factors.”

5. No psychiatric diagnostic interviews to confirm the diagnosis is a significant weakness. Gold standard (structured interview, or semi-structured interview, or a diagnosis confirmed by psychiatrist) should be used to confirm whether GP’s rating depression is under or over-diagnosed. The results of screening scale may also have problems of false negatives and false positives.

We agree with this point. To confirm the GP’s depression rating or the PHQ-9 result a diagnostic interview should have been performed. We mentioned this point in our limitation section. Due to limited resources we were not able to perform a standardized interview. However, we believe that the combination of both, the PHQ-9 and the GP rating reduces the proportion of false-positives and negatives. Future studies should compare the results with a reference standard. Based on your comment, we try to emphasize in the limitations section that the false-negatives and positives are difficult to determine without a gold standard: 

“As it is difficult to determine the accurate proportion of false-positives and false-negatives without a reference standard, further studies need to investigate the accuracy of GP diagnoses in combination with screening questionnaires compared to standardized diagnostic interviews as a reference standard”.

6. The results of PHQ-9 was based on scores (10-14; 15-19; 20- 27) instead of using the German version of the PHQ-9 cut-point or the gold standard for diagnostic interviews. According to a meta-analysis in 2021 (J Affect Disord. 2021 Jan 15;279:473-483.), the accuracy of the PHQ-9 was evaluated in 31 (74%) studies with a two-stage screening system, with structured interview most often carried out by primary care and mental health professionals. Most of the studies employed a cut-off score of 10 (N=24, 57%; total range 5-15). The authors may cite this paper to support why they used PHQ-9 scores (10-14; 15-19; 20- 27) instead of using the German version of the PHQ-9 cut-point. Also, they need to consider the false negatives and false positives.

We are sorry for our misleading presentation. The point you raised is true and we agree with you that a cut-off score of 10 is the ideal way for categorization, which we also used in our study. To avoid confusion for the reader, we deleted “In patients with major depressive symptoms, a score of 10 and higher can be expected, with moderate (10-14), distinct (15-19) and most severe (20-27) levels of the disorder.” Further on, we explain the rationale of the cut-off ≥10 more in detail now. Now we write: 

“The depression severity score comprises nine items which can be summarized, with a range from zero (no depression) to 27 (maximum). Findings from previous studies show that the use of a cut-off value of 10 or higher is considered useful [40] as a score of 10 represents at least a moderate level of depressive symptoms [39]. A score between five and 10 is mostly found in patients with mild or subthreshold depressive symptoms and corresponds to a mild degree of severity [39]".

We adapted the methods section to explain that analysis by scores was used as an additional analysis to the main analysis, in which we focused on the cut-off value when comparing the discordant and concordant subgroups. However, we used the PHQ-9 scores for further analysis and to examine the reliability of the results.

“For this purpose, a PHQ-9 ≥10 was used to indicate a self-rated depression. This outcome was labelled as PHQ-9 positive, PHQ-9 <10 was labelled as PHQ-9 negative. Two groups with concordant versus discordant PHQ-9 and GP ratings at t1 were defined. The replicability of PHQ-9 results between t1 and t2 was compared between the concordant and discordant group by a 2 x 2 table. Mean values were presented additionally to enable a comparison of the dimensional PHQ-9 results. Furthermore, reliability of the PHQ-9 test results at t1 and t2 was assessed within these groups and within the entire sample by Cohen’s Kappa, Pearson’s correlation coefficient and Bland-Altman plots”.

In the absence of a structured interview, it is hardly possible to take false-positives and false-negatives into account in our analyses. We do, however, discuss this limitation of the present study in the limitations section. In our discussion we added the point that the false-positive rate of the PHQ-9 is around 60% and discussed it in the context of our results: 

“Studies have shown that the false-positive rate of the PHQ-9 is around 60% in a population with a 10% prevalence of depression so that false-positives and false-negatives have to be examined carefully [30]. Overall we found that 46.7% of the patients with a PHQ-9 positive result at t1 had a negative result at t2 in the present study. This may indicate a significant overestimation of PHQ-9 positives at t1. In the discordant subgroup, this was even more expressed (60.0%) than in the concordant subgroup (37.0%). Therefore, agreement in the GP and PHQ-9 assessments may be associated with a decreased likelihood of false-positives.”

In the discussion: 

7. Patients who were not rated as depression by their GP had less PHQ positive results at T1 and T2. In fact, Among 20 patients (36%) with PHQ-9 positive at T1 in the discordant group (n=55), a significant proportion (40%) of patients received a positive results at T2 again, which implies the tendency of chronicity.

Thank you for raising this interesting point, we have included this aspect in the discussion. The tendency of chronicity of depression has been shown in several studies. However, we see this tendency of chronicity of depression even more in the concordant subgroup (63.0%) which is why the GP assessment needs to be standardized and evaluated and then taken into account in the diagnostic process consistently. The consideration of the GP assessment could lead to a better diagnostic process and treatment for patients with severe depression with a high risk of chronicity. Now we write: 

“Furthermore, it has to be taken into account that among 20 patients with PHQ-9 positive results at t1 in the discordant subgroup, a significant proportion (40.0%) of patients received a positive result at t2 again, which implies the general tendency of chronicity [41, 42]. Nevertheless, there is a remarkable difference to the concordant subgroup where chronicity rates are higher (63.0%). This might indicate that a combination of GP heuristics and screening questionnaire could improve the diagnostic process of patients with severe depression and a high risk of chronicity, which needs to be investigated in further studies.”

---

## [Decision Letter · Decision Letter 1]

10 Oct 2022

Combining the GP’s assessment and the PHQ-9 questionnaire leads to more reliable and clinically relevant diagnoses in primary care

PONE-D-21-39064R1

Dear Dr. Teusen,

We’re pleased to inform you that your manuscript has been judged scientifically suitable for publication and will be formally accepted for publication once it meets all outstanding technical requirements.

Kind regards,

Pedro Vieira da Silva Magalhaes, M.D., Ph.D.

Academic Editor

PLOS ONE

Additional Editor Comments (optional):

Reviewers' comments:

Reviewer's Responses to Questions

**Comments to the Author**

1. If the authors have adequately addressed your comments raised in a previous round of review and you feel that this manuscript is now acceptable for publication, you may indicate that here to bypass the “Comments to the Author” section, enter your conflict of interest statement in the “Confidential to Editor” section, and submit your "Accept" recommendation.

Reviewer #1: All comments have been addressed

Reviewer #3: All comments have been addressed

2. Is the manuscript technically sound, and do the data support the conclusions?

Reviewer #1: Yes

Reviewer #3: Yes

3. Has the statistical analysis been performed appropriately and rigorously? 

Reviewer #1: Yes

Reviewer #3: Yes

4. Have the authors made all data underlying the findings in their manuscript fully available?

Reviewer #1: Yes

Reviewer #3: Yes

5. Is the manuscript presented in an intelligible fashion and written in standard English?

Reviewer #1: Yes

Reviewer #3: Yes

6. Review Comments to the Author

Reviewer #1: (No Response)

Reviewer #3: I Intersting stuyd with rouboust methodology and desgin and results, insering Bland-Altmann plot was very useful .

congratulation

7. PLOS authors have the option to publish the peer review history of their article (what does this mean?). If published, this will include your full peer review and any attached files.

Reviewer #1: No

Reviewer #3: No

---

## [Editor Report · Acceptance letter]

14 Oct 2022

PONE-D-21-39064R1 

Combining the GP’s assessment and the PHQ-9 questionnaire leads to more reliable and clinically relevant diagnoses in primary care 

Dear Dr. Teusen:

I'm pleased to inform you that your manuscript has been deemed suitable for publication in PLOS ONE. Congratulations! Your manuscript is now with our production department. 

Kind regards, 

on behalf of

Professor Pedro Vieira da Silva Magalhaes 

Academic Editor

PLOS ONE